# Attention-Guided Instance Segmentation for Group-Raised Pigs

**DOI:** 10.3390/ani13132181

**Published:** 2023-07-03

**Authors:** Zhiwei Hu, Hua Yang, Hongwen Yan

**Affiliations:** College of Information Science and Engineering, Shanxi Agricultural University, Jinzhong 030801, China; yanghua@sxau.edu.cn (H.Y.); yhwhxh@126.com (H.Y.)

**Keywords:** image segmentation, feature pyramid network, attention mechanism, channel attention, spatial attention

## Abstract

**Simple Summary:**

In this study, we propose a grouped attention module that combines channel attention and spatial attention simultaneously and applies it to a feature pyramid network for instance segmentation of group-raised pigs. First, we discuss the performance impact of adding different attention modules and setting different numbers of attention groups on the pig instance segmentation. Then, we visualize the spatial attention information and analyze the segmentation results under different scenes, ages, and time periods. Additionally, we explore the robustness and transferability of the model using third-party datasets. The aim is to provide insights into the intelligent management of pigs.

**Abstract:**

In the pig farming environment, complex factors such as pig adhesion, occlusion, and changes in body posture pose significant challenges for segmenting multiple target pigs. To address these challenges, this study collected video data using a horizontal angle of view and a non-fixed lens. Specifically, a total of 45 pigs aged 20–105 days in 8 pens were selected as research subjects, resulting in 1917 labeled images. These images were divided into 959 for training, 192 for validation, and 766 for testing. The grouped attention module was employed in the feature pyramid network to fuse the feature maps from deep and shallow layers. The grouped attention module consists of a channel attention branch and a spatial attention branch. The channel attention branch effectively models dependencies between channels to enhance feature mapping between related channels and improve semantic feature representation. The spatial attention branch establishes pixel-level dependencies by applying the response values of all pixels in a single-channel feature map to the target pixel. It further guides the original feature map to filter spatial location information and generate context-related outputs. The grouped attention, along with data augmentation strategies, was incorporated into the Mask R-CNN and Cascade Mask R-CNN task networks to explore their impact on pig segmentation. The experiments showed that introducing data augmentation strategies improved the segmentation performance of the model to a certain extent. Taking Mask-RCNN as an example, under the same experimental conditions, the introduction of data augmentation strategies resulted in improvements of 1.5%, 0.7%, 0.4%, and 0.5% in metrics *AP*^50^, *AP*^75^, *AP*^L^, and *AP*, respectively. Furthermore, our grouped attention module achieved the best performance. For example, compared to the existing attention module CBAM, taking Mask R-CNN as an example, in terms of the metric *AP*^50^, *AP*^75^, *AP*^L^, and *AP*, the grouped attention outperformed 1.0%, 0.3%, 1.1%, and 1.2%, respectively. We further studied the impact of the number of groups in the grouped attention on the final segmentation results. Additionally, visualizations of predictions on third-party data collected using a top-down data acquisition method, which was not involved in the model training, demonstrated that the proposed model in this paper still achieved good segmentation results, proving the transferability and robustness of the grouped attention. Through comprehensive analysis, we found that grouped attention is beneficial for achieving high-precision segmentation of individual pigs in different scenes, ages, and time periods. The research results can provide references for subsequent applications such as pig identification and behavior analysis in mobile settings.

## 1. Introduction

With increasing concerns about food safety and health, the pork industry is placing greater emphasis on pig breeding and management. However, outbreaks of contagious diseases such as African swine fever and increasing environmental pressures have posed many challenges to the pig farming industry [1]. To improve pig farming efficiency and health, monitoring and managing the health status of pigs has become crucial. Traditional methods of monitoring pig health mainly rely on manual observation, but this approach is not only time-consuming and labor-intensive but also susceptible to subjective factors and has a certain misjudgment rate. As pig farming is moving towards intensification, scale, and facility-based management, fine pig farming based on individual management and quality assurance that meets animal welfare requirements has become the trend in the pig farming industry. Instance segmentation technology can automatically analyze and recognize various features of pigs by capturing video or image data of pigs in their natural state and providing diagnosis results [2]. This method not only reduces labor costs but can also more accurately identify and diagnose the health status of pigs, thereby improving pig farming efficiency and health.

Convolutional Neural Network (CNN) has a powerful feature extraction ability for images, and it has been widely used in pig behavior identification [1,3], pig face recognition [4,5,6,7,8], pig multi-target detection [9,10], pig counting [11], pig detection [12,13,14], and other fields [15,16]. In the field of pig image segmentation, for locating sow image segmentation from the overhead views of commercial pens, Yang [17] first used a fully convolutional network (FCN) to segment the pig, then refined the coarse output of the FCN using the probability map from the final layer of the FCN and Otsu’s thresholding from the hue, saturation, and value color information. For automatic recognition of nursing interactions under commercial farm conditions by using spatial and temporal information of nursing behavior, Yang [18] used an FCN module to segment the sowing part, and then the udder zones were calculated dynamically by the geometrical properties of the nursing sow and the piglet length. Yang [19] developed a method to automatically recognize nursing behaviors in videos by exploiting spatiotemporal relations and FCN-based semantic segmentation. However, the above CNN-based segmentation methods have the following shortcoming: they only separate pig individuals from the breeding environment and cannot distinguish different pig individuals in the same image, while instance-level segmentation can be used to discriminate different individuals in the same class, which is more suitable for downstream tasks such as pig identification.

Many scholars have conducted exploratory research on instance segmentation tasks, such as the instance segmentation of cattle [20], weed [21], and other fields [22,23]. In terms of pig instance segmentation, Tu [24] explored the Mask Scoring R-CNN (MS R-CNN) to segment adhesive pig areas in group-pig images to avoid target pigs being missed and error detection in overlapping or stuck areas of group-housed pigs, using the soft non-maximum suppression (soft-NMS) by replacing the traditional NMS. Gan [25] employed an anchor-free deep learning network for instance segmentation of individual sows and piglets. Specifically, they used the attention graph convolution-based structure to distill element-wise features and further applied the mask head for the mask prediction. Brünger [26] defined the pig panoptic instance segmentation task and presented different network heads and postprocessing methods to aim at the pixel-accurate segmentation of the individual pigs. However, the above-mentioned pig instance segmentation method only applies or modifies Mask R-CNN simply, and does not perform substantial model structure adjustment. The pig nose, pig ears, and other parts are rich in biological information, and effectively distinguishing them can further improve segmentation precision. The attention mechanism can be used for discriminating feature selection. When extracting information, it can strengthen the weight of regional features related to the task and improve the task effect. It has the advantage of being plug-and-play and is easy to embed in various task models. It has achieved good results in many fields and many scholars also use this mechanism in the field of pig research. Liu [27] used the idea of recurrent residual attention mechanisms in the feature pyramid network, applied Mask R-CNN and Cascade Mask R-CNN as the task networks, with ResNet50 and ResNet101 as the backbone networks, and further discussed the model under different combination modes on the instance segmentation performance of group-raised pigs. Hu [2] proposed to embed the channel attention and spatial attention information into the feature pyramid network for instance segmentation of group-raised pigs, and used Mask R-CNN, MS R-CNN, Cascade Mask R-CNN, and HTC as the task network, ResNet50 and ResNet101 as the backbone network, and verified individual attention performances under different combinations of conditions. However, the above-mentioned studies on pigs based on attention mechanisms directly embed channel or spatial attention information into the feature pyramid structure, which mainly has the following three shortcomings: (1) *They only use the inherent hierarchical structure of the feature pyramid, ignoring the hierarchical information of different channels within the layer feature map;* (2) *They share the same channel or spatial attention structure among the internal channels of the feature map and do not perform deep-level differentiated attention extraction;* (3) *They all use the entire feature map for attention acquisition, which is inefficient and computationally intensive.* ShuffleNet [28] and EPSANet [29] achieve hierarchical parallel extraction of multi-channel information by grouping feature maps to build a multi-branch structure and have achieved good results. Inspired by these works, in order to achieve hierarchical, fine-grained, and efficient attention extraction, we group the feature maps according to the channel and filter the channel and spatial attention information on the feature maps within the group, respectively. At the same time, the group attention module is embedded in the feature pyramid network to realize the instance segmentation of group-raised pigs.

By grouping the feature maps, we extract the channel and spatial information in the group to construct the grouped attention module and embed it into the pyramid network to realize non-contact and damage-free instance segmentation of group-raised pigs in scenes such as deep separation, high adhesion, pigpen occlusion, and different age stages. Our attention module can achieve multi-scene pig individual instance segmentation and provide technical support for downstream tasks such as pig identification. The main contributions include the following aspects, and the terms used in this paper are listed in Table 1:Taking ResNet50 as the backbone network and Mask R-CNN, and Cascade Mask R-CNN as the task networks, we compared the data augmentation operations during training and attention modules (1111 and 0010) to the backbone to explore the impact of two strategies on pig instance segmentation.Compared with the existing attention modules CBAM, DANet, and SCSE, it is proven that our proposed GAM attention module is more effective.We explore the influence of the number of groups on the performance of group attention to select the optimal number of groups.Model tests were performed on the datasets with different adhesion degrees, different ages, and third-party datasets that were not participating in training to verify the robustness and transferability of our model.

## 2. Materials and Methods

### 2.1. Data Source

The experimental data were collected from Jicun Town, Fenyang City, Shanxi Province, China, which was denoted as Farm One, and the Laboratory Animal Management Center of the Shanxi Agricultural University of China, which was marked as Farm Two. The corresponding collection time, collection temperature, collection environment, and pigpen size of each pig farm are shown in Table 2. We used a Canon 700D anti-shake camera to shoot, and the collection time of each pigpen was more than 60 min to meet the continuity of the data itself. We selected Large White, Landrace, and Duroc breeds of pigs as the research objects. The age range of the pigs was 20~105 d, and each pigpen contained 3~8 individuals. Each pig farm selected 4 pigpens as the experimental objects, and finally, a total of 45 pigs were obtained for model training and testing.

### 2.2. Data Collection

Traditional studies on individual pigs mostly use the top-mounted method for data collection [1,2,10,24]. Different from this, we used a horizontal viewing angle and a non-fixed lens for video shooting; take Pig Farm Two as an example. The schematic diagram of the experimental data collection platform is shown in Figure 1. Compared with the top-down view, the head-up perspective has unique academic and applied value as follows:The camera lens used for collection can be conveniently rotated and retracted, which is conducive to capturing data under changing conditions;This method can capture the face, hoof, and other areas with rich biological information to a greater extent, which is conducive to the identification of live pigs;Behavioral data such as climbing and aggression can be collected from a human perspective, which is helpful for pig behavior identification.

### 2.3. Data Preprocessing

The following operations were performed on the collected videos to obtain the instance segmentation dataset of group pigs:The collected videos were cut every 25 frames, and the resolution size of the obtained picture was 1920 × 1080, which was adjusted to 2048 × 1024 according to the aspect ratio of 2:1. The edge area was filled with white pixels, and LabelMe (http://labelme.csail.mit.edu/Release3.0/) (accessed on 5 April 2023) was used for data labeling. In order to reduce the memory usage of the model, the overall annotation data was scaled to 512 × 256, and finally, 1917 images were obtained as the initial annotation dataset, which was divided into 959 training sets, 192 validation sets, and 766 test sets.Then, the data augmentation operation performed during training was performed on the initially labeled dataset. Different from traditional related studies that preprocess the augmented data [1,3,11,25], the advantages of using data augmentation during training are as follows: (a) the generated data is more random; (b) each image has a probabilistic enhancement operation in each iteration process, avoiding the pre-enhancement strategy to limit the diversity of image data. The methods and their corresponding parameters used for data augmentation during training are shown in Table 3.

## 3. Pig Instance Segmentation Model

### 3.1. Mask R-CNN and Cascade Mask R-CNN Task Model

Mask R-CNN [30] and Cascade Mask R-CNN [31] are both composed of backbones (such as ResNet, Region Proposal Network (RPN) [32], Feature Pyramid Network (FPN), ROIAlign, and functional output). In order to avoid the performance loss problem of a single head network due to improper setting of the Intersection over the Union (IOU) threshold, Cascade Mask R-CNN continuously refines the segmentation results by concatenating multiple IOU threshold head networks, as shown in ①~③ in Figure 2. The IOU thresholds of ①, ②, and ③ were set to 0.5, 0.6, and 0.7, respectively.

### 3.2. Feature Pyramid Network after Adding Group Attention

Convolutional networks can be used to extract features from different angles, hierarchically. The deep-level features are rich in semantics but lack location information, while the shallow-level features pay more attention to location information but lack semantic content. The Feature Pyramid Network (FPN) hierarchically fuses deep and shallow information through skip connections, which can be used to solve the inconsistency between semantics and location information in the process of feature map fusion. The FPN structure is shown in Figure 3. It consists of two stages: bottom-up and top-down. For the bottom-up stage, after each 0.5× down-sampling operation, the corresponding feature map resolution is reduced to half of the original size. For the top-down stage, after each 2× up-sampling operation, the resolution size of the corresponding feature map is expanded to twice the original size. In stages bottom-up and top-down, feature maps of the same resolution are directly added to obtain the corresponding level output. Traditional FPN only performs simple bit-wise superposition fusion of deep and shallow features at the same level, ignoring the nonlinear dependencies between feature maps of different depths. The attention mechanism can apply differential weights to the information in the feature map and selectively activate task-related regions to improve the feature fusion performance. Group Attention (GA) can strengthen the regions with high task relevance and weaken the task-irrelevant regions without significantly increasing the number of parameters. Inspired by ResNeSt and SA-Net, we innovatively introduce the idea of group attention into the fusion of deep and shallow features at each level of FPN. The structure can be seen in the shaded part of Figure 3. The Group Attention Module (GAM) consists of three units: Group Split Unit (GSU), Attention Selection Unit (ASU), and Content Aggregation Unit (CAU).

Note that the green shaded area represents the Mask R-CNN model. ‘I’ means the input image, backbone means the backbone network, FPN means feature pyramid network, RPN represents the region proposal network, Align means ROIAlign, H0~H3 means four different head networks, C0~C3 mean the classification results, M0~M3 mean segmentation results, and B0~B3 indicate the results of the detection box.

**Group Split Unit (GSU):** For the linear superposition, result **X** of the deep feature map $F_{high}$ and the shallow feature map $F_{low}$ were split into n feature maps **G** with the same number of channels along the channel dimension, satisfy $X=[G_{1},G_{2},\ldots ,G_{n}]$, where $X\in R^{C\times H\times W}$, $G_{i}\in R^{\bar{C}\times H\times W},i=1,2,\ldots ,n,\bar{C}=\frac{C}{n}$, *C*, *H*, *W* represent the number of channel, height, and width of the feature map, respectively, and n is selected as the power of 2. Taking into account the complexity of the model and the degree of performance improvement, n is set to 8, 16, 32, and 64 for experiments to explore the most suitable number of groups. In addition, if not explicitly stated, the number of groups in the GAM module is set to 16.

**Attention Selection Unit (ASU):** We performed two convolution operations on the grouped feature map $G_{i}$ with a number of channels $\bar{C}$ and a convolution kernel size of 3 × 3 to generate **Bc** and **Bs**, as the inputs of the Channel Attention Branch (CAB) and Spatial Attention Branch (SAB), respectively. CAB can be used to encode the mapping relationship between channels to generate a channel-enhanced feature map $F_{cab}$, and SAB can be used to guide feature maps to perform pixel-level content selection to generate a dense position-enhanced feature map $F_{sab}$. Finally, we linearly superimposed the channel and spatial attention selection results to obtain the attention-enhanced output $F_{asu}$.

**Content Aggregation Unit (CAU):** We spliced the feature map $G_{i}$ filtered by the attention of different groups along the channel dimension to obtain the final output **Y**, which satisfies $Y=[P_{1},P_{2},\ldots ,P_{n}]$, $Y\in R^{C\times H\times W}$, $P_{i}\in R^{\bar{C}\times H\times W},i=1,2,\ldots ,n,\bar{C}=\frac{C}{n}$. It should be noted that the output feature map **Y** has the same dimension as the input feature map **X** so that our group attention module can be easily inserted into any existing model, which shows that our module has plug-and-play properties.

Note that the shaded part represents the group attention module. Conv represents the convolution operation. T1~T5 represent the different bottom-up layers. M2~M5 represent the different top-down layers. G1~Gn represent the different group split units. C1~Cn represent the different content aggregation units. Bc and Bs denote feature mps for channel attention branch and spatial attention branch, respectively.

### 3.3. Channel Attention Branch

Different channels in the feature map can be regarded as responses to a specific category, and different semantic responses are related to each other. In order to fully model the dependencies between channels to strengthen the feature mapping between related channels and improve the semantic feature representation, inspired by PSA [33], we build a channel attention branch in each grouping to explicitly encode channel dependencies. The structure is shown in Figure 4.

The CAB branch performed the following three-step operations on the input feature map I to obtain the output feature map O, recalibrated by channel attention:(1)We performed a convolution operation on the input feature map **I** with a convolution kernel size of 3 × 3, and the number of channels set to C and 1, to obtain feature map **B** and channel-compressed feature map **A**, respectively. Where $I\in R^{C\times H\times W}$, $A\in R^{1\times H\times W}$, and $B\in R^{C\times H\times W}$.
(2)Firstly, we performed dimension transformation and transposition operations on feature map **A** to obtain feature map $\tilde{A}$, and also performed dimension transformation on feature map **B** to obtain feature map $\tilde{B}$, where $\tilde{A}\in R^{HW\times 1\times 1}$, $\tilde{B}\in R^{C\times HW}$. Then, we performed matrix multiplication of $\tilde{B}$ and $\tilde{A}$, at the same time, the softmax activation function was used to process the multiplication result to obtain the channel attention map **X**, where $X\in R^{C\times 1\times 1}$. The value $x_{i}$ of each element in **X** represents the attention value of the *i*-th channel after the action of the remaining channels, and its value can be calculated by Formula (1), where ${\tilde{B}}_{i}$ represents the value of the *i*-th row of $\tilde{B}$ and exp(*) represents the exponential function with base e.



(1)
$$x_{i}=\frac{exp\left({\tilde{B}}_{i}\cdot \tilde{A}\right)}{\sum_{t=1}^{C}exp\left({\tilde{B}}_{t}\cdot \tilde{A}\right)}$$



(3)We bit-wise multiplied the channel attention map **X** with the input **I** to get the channel attention-enhanced representation output **O**, where $O\in R^{C\times H\times W}$.

### 3.4. Spatial Attention Branch

Unlike channel attention, which treats different location information in the same channel equally, spatial attention can be used to encode pixel-level dependencies, which can be used to make up for the shortcomings of channel attention information filtering. Inspired by PSA and CBAM [34], we introduced the Spatial Attention Branch (SAB), which uses the response values of all pixels in a single channel feature map to the target pixel to build pixel-level dependencies and further guided the original feature map to filter spatial location information to generate contextual output. The structure can be seen in Figure 5.

The SAB branch performed the following three-step operations on the input feature map **I** to obtain the feature map **O** recalibrated by spatial attention:(1)We performed a two-way convolution operation on the input feature map I with a kernel size 3 × 3 and the number of channels $\frac{C}{2}$, and obtain two feature maps **A** and **B** with the same size, where $A\in R^{C/2\times H\times W}$, and $B\in R^{C/2\times H\times W}$. The feature map **A** was compressed in the spatial dimension by two methods, Global Average Pooling (GAP) and Global Max Pooling (GMP), and the two compression results are superimposed bitwise to obtain the feature map $\hat{A}\in R^{C/2\times 1\times 1}$. At the same time, the feature map **B** was dimensionally transformed to obtain the feature map $\tilde{B}$.
(2)We first performed dimension transformation and transposition operation on feature map $\hat{A}$ to obtain feature map $\tilde{A}$, and then performed matrix multiplication operation on $\tilde{A}$ and feature map $\tilde{B}$. Then, the operation result was dimensionally transformed to obtain $X\in R^{1\times H\times W}$, and the softmax activation function was spliced to obtain the spatial attention weight map $Y\in R^{1\times H\times W}$. Each element $x_{i,j}$ in **X** and $y_{i,j}$ in **Y** represents the activation value at the position of the *i*-th row and the *j*-th column in the corresponding feature map, which can be represented by Formula (2), where the two terms in the formula get the superimposed activation values from the column and row angles respectively.



(2)
$$y_{i,j}=\frac{exp\left(x_{i,j}\right)}{\sum_{r=1}^{H}exp\left(x_{r,j}\right)}+\frac{exp\left(x_{i,j}\right)}{\sum_{c=1}^{W}exp\left(x_{i,c}\right)}$$



(3)We multiplied the spatial attention map **Y** by the input **I** to obtain the output $O\in R^{C\times H\times W}$ filtered by spatial attention.

## 4. Experiment

### 4.1. Implementation Details

The experimental platform is 4 16-G Tesla P100 GPUs, and the code is written using the mmdetection (https://github.com/open-mmlab/mmdetection) (accessed on 5 April 2023) framework. All models set the batch size to 8 and the number of iterations to 10. We adopt Stochastic Gradient Descent (SGD) as the optimizer, the initial learning rate is set to 0.02, the momentum magnitude is set to 0.9, and the regularization weight decay coefficient is set to 0.0001. Consistent with the default settings of mmdetection, the three-channel values of the image are normalized with a mean of (123.675, 116.28, 103.53) and a standard deviation of (58.395, 57.12, 57.375).

### 4.2. Evaluation Metrics

We chose average precision (*AP*) as the evaluation standard to measure the performance of the model for instance segmentation of group-raised pigs. *AP* represents the area under the Precision-Recall curve, which can be shown in Formulas (3)–(5), where *TP* (True Positive) represents the number of pixels correctly predicted as the pig category, *FP* (False Positive) denotes the number of pixels wrongly predicted as the pig category, and *FN* (False negative) represents the number of pixels predicted as the background instead of the pig category. Similar to COCO (https://cocodataset.org/) (accessed on 5 April 2023), three IOU thresholds of 0.5, 0.75, 0.5~0.95:0.05 (where 0.05 represents the growth step) were selected to measure the model segmentation performance under different conditions, which were recorded as *AP*^50^, *AP*^75^, *AP*. At the same time, based on the individual size of the pigs, we divided them into small targets (the area of individual pigs < 322), medium targets (322 < area of individual pigs < 962), and large targets (the area of individual pigs > 962). In particular, we separately calculated the AP index value under the large target with IOU value between 0.5 and 0.95, which was denoted as *AP*^L^.
(3)$$Precision=\frac{TP}{TP+FP}$$
(4)$$Recall=\frac{TP}{TP+FN}$$
(5)$$AP=\int_{0}^{1}Precision\cdot Recall\;dr$$

### 4.3. Main Results

#### 4.3.1. The Effects of Backbone Attention, Data Augmentation, and Deformable Convolution

In order to explore the impact of data augmentation strategies on model segmentation results during training under different task network conditions, we chose ResNet50 as the backbone network, Mask R-CNN, and Cascade Mask R-CNN as the task networks to explore the segmentation performance before and after data augmentation operations are introduced during training; In order to explore whether adding an attention module to the backbone is beneficial to feature extraction, two attention structures, 0010 and 1111 [35], were selected and embedded in the Stages 3 and 4 of the ResNet50 backbone network. In backbone networks, attention mechanisms can also be introduced, and the performance of the attention mechanism can be influenced by several factors, including the content of the query and key, the content of the query and relative position, only the content of the key, and only the relative position. Among them, attention formed by simultaneously using all four types of content is referred to as “1111”, while attention that only incorporates the content of key is referred to as “0010”. We conduct the comparative experiment under various task network conditions. In order to avoid the spatial deformation of individual pig images caused by uncontrollable factors in the pig breeding environment and to solve the segmentation problems caused by changes in shooting perspective and pig posture, we introduced a deformable convolutional network (DCN) in stages 2~4 of the ResNet50 backbone network, which can learn spatially dense geometric deformation. DCN is an improved operation on the basis of traditional CNNs, by making small offsets to the sampling positions, it enables non-linear deformation modeling of input features. In image segmentation, it can provide more accurate position and shape information. We replaced the regular convolutions in ResNet50 with DCN to improve the accuracy of the segmentation results. The experimental results of the AP index values corresponding to each condition are shown in Table 4.

***Select different backbone attention:*** The introduction of 0010 and 1111 attention blocks in stages 3 and 4 of RestNet50 shows different performance and better AP index value can be obtained after the introduction of 1111. Taking the Mask R-CNN task network as an example, without introducing Albu and DCN, the *AP*^50^, *AP*^75^, *AP*^L^, and *AP* indicators reached 91.6%, 82.0%, 69.6%, and 67.2%, respectively after adding 1111, which is 1.7%, 1.8%, 1.2%, and 1.2% higher than adding 0010 in each indicator. After introducing Albu strategy and DCN, compared with adding 0010 module, adding 1111 increases *AP*^50^, *AP*^75^, *AP*^L^, and *AP* indicators by 0.4%, 1.3%, 0.9%, and 0.8%, respectively, and the performance improvement is reduced. The Cascade Mask R-CNN task network has a similar pattern.

***Before and after data augmentation during training:*** Introducing the data augmentation strategy during training can improve the segmentation performance of the model under the same conditions to a certain extent. Taking the task network Mask R-CNN with 0010 backbone attention as an example, before the introduction of Albu, the *AP*^50^, *AP*^75^, *AP*^L^, and *AP* index values reached 89.9%, 80.2%, 68.4%, and 66.0%, respectively. After the introduction of Albu, the indicators increased of 1.5%, 0.7%, 0.4%, and 0.5%, respectively. Under other conditions, the introduction of Albu can bring different degrees of index value improvement. It shows that the data augmentation strategy during training helps the model to learn more knowledge and facilitate its subsequent prediction tasks. The main reason is that the image can be adjusted by translation, brightness, contrast, etc. During training to obtain more diverse data so that the model can perceive a variety of scene information and improve the robustness of the model. In addition, from Table 4, we can see that the performance improvement before and after the introduction of Albu is relatively limited, indicating that our initial training dataset already contains rich and extensive data.

***Before and after the introduction of deformable convolution:*** Adding deformable convolution DCN can further improve the value of each AP index based on the introduction of data enhancement Albu during training. Take the task network Mask R-CNN with the addition of 1111-Albu (representing the introduction of 1111 attention in the ResNet50 backbone network and the use of training-time data augmentation strategy) as an example. After further using DCN in ResNet50, the four indicators of *AP*^50^, *AP*^75^, *AP*^L^, and *AP* increased by 0.3%, 0.5%, 0.5%, and 0.4%, respectively. Additionally, it shows a similar trend on the Cascade Mask R-CNN task network that has used the Albu strategy and added 0010 or 1111. The main reason is that the regular lattice sampling in standard convolution makes it difficult for the network to adapt to geometric deformation. After the DCN strategy is adopted, the convolution kernel can be expanded in multiple directions, changing the range of the receptive field so that it is no longer limited to a rectangular area, which is highly consistent with the individual shape of pigs, such as pig trotters, pig nose, and other parts that are not regular graphics. The use of deformed convolution can refine the segmentation of such irregular pig body parts and further improve segmentation accuracy.

#### 4.3.2. Add Different Attention Modules

In order to explore the prediction results of task network segmentation by different attention modules to find the most suitable attention for the pig instance segmentation task, under the same experimental conditions, CBAM, DANet [36], and SCSE [37], which also have channel and spatial attention information, are used to replace the GAM module for prediction on the test set. The prediction results for each AP value are shown in Table 5. For the CBAM module, use Channel Attention Module (CAM) and Spatial Attention Module (SAM) to replace the CAB and SAB in Figure 3; For DANet, use Channel Attention Module (CAM) and Position Attention Module (PAM) to replace CAB and SAB in Figure 3, respectively; For SCSE, use Spatial Squeeze and Channel Excitation (cSE) and Channel Squeeze and Spatial Excitation (sSE) to replace CAB and SAB in Figure 3, respectively.

***Different attention modules:*** For the same task network, after adding different attention modules, the *AP* values of each segmentation result are different. For the baseline attention modules CBAM, DANet, and SCSE, when different task networks are selected, the stability of each attention module is not strong in terms of each AP index value. For example, for the Cascade Mask R-CNN task network added to 0010-Albu-DCN, CBAM performs the best on the *AP*^50^ indicator, but for the *AP*^L^ and *AP* indicators, SCSE performs better, which shows that the existing three attention modules can only achieve the best experimental results on specific indicators on some specific models, and the stability of corresponding attention modules may be affected when the model is switched or the experimental conditions are changed. Compared with CBAM, DANet, SCSE, and other modules, the group attention GAM can achieve better AP index values. Taking Mask R-CNN with 1111-Albu-DCN as an example, using GAM module improves the *AP*^50^, *AP*^75^, *AP*^L^, and *AP* indicators by 1.0%, 0.3%, 1.1%, and 1.2%, respectively, compared with CBAM. Compared with DANet, the improvement is 0.6%, 0.7%, 1.0%, and 1.1%, respectively, and 0.7%, 0.7%, 0.6%, and 0.7%, respectively, when compared with SCSE. This shows that the attention content obtained after grouping the feature maps is more accurate for the instance segmentation results of group-raised pigs. The main reason lies in the fact that by grouping the feature maps, the feature maps are layered to a certain extent, and by performing channel and spatial attention screening on the feature maps within the group, the channel attention selectivity increases the number of channels within the group and improves the prediction accuracy of pig categories. Spatial attention assigns fine-grained weights to intra-group feature maps from the position dimension and strengthens the activation values of regions such as pig noses and pig ears. The feature maps for different groups do not share attention weights. The feature maps between groups are filtered and aggregated after their respective attention information to further strengthen the regional information that is conducive to the segmentation of pig instances and weaken the influence of complex factors such as background and light intensity.

#### 4.3.3. Set Different Number Groups

In order to explore the influence of the number of groups on the performance of group attention, under the same experimental conditions, the number of groups was set to 8, 16, 32, and 64, respectively. Taking the Mask R-CNN and Cascade Mask R-CNN task networks with Albu-1111-DCN information added as an example, the calculated AP index values are shown in Table 6.

***Different number of groups:*** Using group attention modules with different numbers of groups, the task network performance is quite different. Under the same experimental conditions, with the increase in the number of groups, the corresponding AP index values oscillated, basically showing a trend of first increasing and then decreasing and the best AP value can be obtained when the number of groups is set to 16. For the Mask R-CNN task network, when the number of groups is set to 16, the metric values of *AP*^50^, *AP*^75^, *AP*^L^, and *AP* reach 93.3%, 84.7%, 72.7%, and 70.3%, respectively, which are 1.2%, 1.1%, 1.3%, and 1.4% higher than the number of groups of 64 in each metric. Compared with the number of group 64, the indicators are improved by 0.2~0.5%. It means that it is not the more the number of groups, the more beneficial the GAM module is to the instance segmentation of pigs. Set a smaller number of groups, the number of channels of each grouped feature map is large, because the channel and spatial attention information is shared between the feature maps within the group, and the two kinds of attention information between feature maps between groups are independent. The small number of groups weakens the meaning of grouping so that most feature maps still share the same attention information and the attention weights of feature maps cannot be differentiated. However, when a large number of groups are set, the number of channels of the feature map in each group is small. When obtaining channel and spatial attention based on feature maps, less information is used, which may lead to misjudgment of attention, resulting in each group of feature maps giving more differential weights to the same position. When the feature maps filtered by attention are finally aggregated, there is a phenomenon of erroneous superposition or blurred judgment, which affects the segmentation results. Hence, it is appropriate to set the number of groups to 16.

#### 4.3.4. Changes in the Amount of Parameters and the Amount of Calculation

In deep learning, model parameter count and computational complexity are two important indicators used to measure the complexity of a model and its computational resource requirements. Model parameter count refers to the number of adjustable parameters that need to be learned in the model. These parameters are used to represent the weights and biases of the model, and by adjusting them, the model can adapt to the given training data. A larger parameter count generally indicates a stronger representation capacity of the model, but it also increases the model’s complexity and memory consumption. Computational complexity refers to the total number of computational operations required during model inference or training. Computational complexity is typically related to the model’s structure and the size of the input data. In deep learning, computational complexity is often measured in terms of the number of floating-point operations (FLOPs) or the count of multiplication-addition operations. A higher computational complexity implies the need for more computational resources (such as CPUs or GPUs) to perform the model’s forward and backward propagation, thereby increasing the computation time and cost. The parameters and calculation results before and after adding different attention are shown in Table 7. It can be found that the introduction of attention does bring a certain amount of time and space overhead, but this overhead is relatively reasonable and tolerable.

### 4.4. Visualization

#### 4.4.1. Spatial Attention Visualization

In order to more intuitively understand the effectiveness of the attention mechanism, we focus on the content of the spatial attention module. Take the best-performing Cascade Mask R-CNN-Albu-1111-DCN as an example, the output of the T2 layer in the FPN module is extracted as the shallow feature, and the output of the up-sampled M3 layer is used as the deep feature. Part of the feature map filtered by the spatial attention branch is shown in Figure 6, where the brightness corresponds to the activation value, and the brighter the activation value, the larger the activation value.

It can be seen from Figure 6 that the specific semantic long-distance dependency information is significantly enhanced after being filtered by the spatial attention branch. For #1 in the input image, after being screened by SAB, it pays more attention to the external growth environment, such as pigpens and pigsties. For #2, the region where the individual pigs are located is given a larger activation value. Even if the distance between different pigs is far away or they are attached, SAB can still capture this long-distance semantically similar information and distinguish it from the background. For #3, it pays more attention to detailed information such as pig trotters, which is beneficial to enhance the weight of distinguishing parts and improve the precision of segmentation. For #4, it mainly extracts the edge information of pig outlines, and spatial attention can effectively separate highly adhesive pigs to improve the individual discrimination of instance-level pigs. To sum up, introducing the spatial attention branch can aggregate denser and richer contextual dependencies and extract regional information with similar semantic categories.

#### 4.4.2. Visualization of Prediction Results for Different Scenarios and Age Stages

In order to explore the robustness of the model after adding different attention under different conditions, the test set of group-raised pigs is divided into two different test subsets according to the scene and age. For the scene subset, there are three categories: deep adhesion, high separation, and pigpen occlusion. For the age subset, the age range is from 20 to 105 days, and three age groups of pigs of 20 days, 32 days, and 65 days are selected for visual display. We used Cascade Mask R-CNN-Albu-1111-DCN as the basic network to explore its instance segmentation effect in different scenes and age stages after adding CBAM, DANet, SCSE, and GAM; the visualization results are shown in Figure 7.

After adding group attention GAM, the segmentation results are better than other existing attention modules in different scenes and age stages data. For the part numbered ③, the GAM segmentation of the leg information is more precise and complete. The reason lies in the fact that the grouped maps do not share attention information, and the enhanced superposition of the leg information by the feature maps in different groups makes the segmentation more rounded. For parts numbered ④ and ⑤, CBAM and DANet misjudged the pigpen as an individual pig, and compared with SCSE, the GAM module can eliminate the influence of the pigpen debris and extract the real pig area; For the part numbered ⑩, even if the pig leg and the pig body part have been spatially separated, the four attention modules can correctly classify it into the appropriate pig category, and the GAM module is more reasonable for the segmentation of pig legs; For pig individuals that deviate from the lens, the head area is blurred, and the adhesion is more serious (such as the part numbered ⑦ and ⑧). None of the four attention modules can distinguish adhering individuals but can segment them as a whole, and the segmentation of the GAM module is more complete. In the follow-up work, we need to focus on adding such complex conditional sample data so that the model can effectively separate pig individuals in more difficult scenarios.

#### 4.4.3. Visualization of Prediction Results for Other Datasets

To explore the model’s transformer performance, we conducted a transfer test on the dataset proposed by Psota. According to the data collection time, the dataset is divided into daytime and nighttime subsets. It should be noted that, on the one hand, this part of the dataset is obtained by the top-mounted method, which is different from the horizontal view perspective we use. On the other hand, this part of the dataset does not have instance-level annotation information and does not participate in the training of models, so it can be used to test the robustness and transferability. We use Cascade Mask Mask R-CNN-Albu-1111-DCN as the basic networks, after adding CBAM, DANet, SCSE, and GAM attention modules, its prediction results on the daytime and nighttime subsets are shown in Figure 8.

As can be seen from Figure 8, GAM attention outperforms other attention modules on both daytime and nighttime data subsets. For the pigs numbered ①, ②, and ③, the light is extremely dim and it is difficult to distinguish whether it is a pig area even with the naked eye, but the GAM attention can still perform a large degree of segmentation, which proves the strong ability of GAM to extract pigs; For the area numbered ④, it is difficult for other attention to segment it, and some cannot separate the adhesion parts, but the GAM module can effectively separate it. For the areas numbered ⑦ and ⑧, the four kinds of attention failed to separate the adherent individuals, but in contrast, the GAM segmentation was more reasonable and complete; For the area numbered ⑨, none of the four types of attention can segment it. On the one hand, these areas are located at the edge of the image. Compared with the central position, the convolutional network has a weaker ability to extract the edge area. On the other hand, this part is heavily occluded by the pigpen, and the information about the exposed pigs is not obvious. To sum up, even for images obtained by different data collection methods, grouped attention still has a certain degree of transfer ability on the dataset that did not participate in training. In subsequent applications, our model can be used as a basic model for fine-tuning to further improve the performance of pig instance segmentation on new datasets.

## 5. Discussion

From the perspective of model selection, we chose Mask R-CNN and Cascade Mask R-CNN as the task networks for experimentation. On one hand, there are other instance segmentation task networks, such as Mask Scoring R-CNN and HTC, are used in practice. However, the main purpose of this study is to validate the effectiveness of attention mechanisms in instance segmentation tasks for group-raised pigs. In the next step, the attention module proposed in this study can be considered for integration into Mask-Scoring R-CNN and HTC networks. On the other hand, we only considered model accuracy as the evaluation metric in our experiments. In the next phase, we plan to apply attention mechanisms to real-time instance segmentation scenarios for group-raised pigs using models such as SOLO and YOLACT. This will not only improve accuracy but also ensure model speed, making them more suitable for practical production practices.

From a data perspective, this study conducted model testing on third-party datasets to validate the stability and robustness of the models. In the next step, we plan to annotate the third-party datasets to expand the dataset size and collect more nighttime scene data to enrich the coverage of the dataset. Furthermore, we will consider using open-source datasets to encourage more researchers to explore the field of instance segmentation for group-raised pigs.

In addition, GPT-4 and DALL·E models, as two generative models based on the Transformer architecture, can have two potential applications in the field of pig instance segmentation for dataset augmentation. On the one hand, GPT-4 can be utilized for instance-level annotation of pig image data. Since GPT-4 supports multimodal operations, it can provide coarse-grained annotations, which can then be further refined through manual verification. On the other hand, with the help of the DALL·E model, more complex pig datasets can be generated based on natural language descriptions. The utilization of these two technologies is expected to bring innovation to the field of pig instance segmentation.

## 6. Conclusions

In this study, a grouped attention module that fuses channel and spatial attention at the same time is introduced into the feature pyramid network. Taking ResNet50 as the backbone network and Mask R-CNN and Cascade Mask R-CNN as the task networks, we discuss the performance impact of adding different attention modules and setting different number of groups on the instance segmentation of group-raised pigs and visualize the spatial attention information. Furthermore, we analyze the segmentation results under different scenes, ages, and time periods. Finally, we explore the robustness and transferability of the model using third-party datasets. The main conclusions are as follows:(1)Introducing 0010 and 1111 attention, deformable convolution, and training-time data augmentation strategies in the backbone network can improve the prediction performance of the model to a certain extent. Additionally, the 1111 attention can get a better AP metric value than 0010 attention.(2)Under the premise of 1111-Albu-DCN, compared with adding CBAM, DANet, and SCSE attention modules, Mask R-CNN-GAM is 0.3~1.2% higher than Mask R-CNN-CBAM, 0.6~1.1% higher than Mask R-CNN-DANet, and 0.6~0.7% higher than Mask R-CNN-SCSE, indicating that GAM is more conducive to the extraction of attention information.(3)Under the same test conditions, with an increase in the number of groups, the corresponding *AP* metric value oscillates, and when the group size is set to 16, the two task networks have the best performance.(4)The visualization of the feature map of the spatial attention branch shows that spatial attention in GAM can aggregate denser and richer contextual dependencies and extract regional information with similar semantic categories.(5)Compared with the addition of CBAM, DANet, and SCSE attention modules, we perform the prediction on different scenes, different ages, and the third-party daytime and nighttime sub-datasets. After adding GAM attention, the segmentation is more complete and finer, indicating that GAM is more robust and has better migration ability.

## Figures and Tables

**Figure 1 animals-13-02181-f001:**
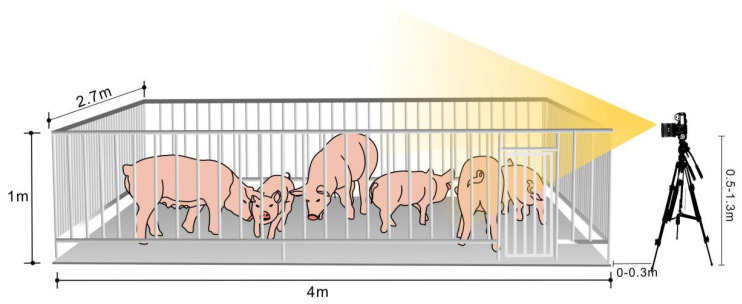
The platform of data collection.

**Figure 2 animals-13-02181-f002:**
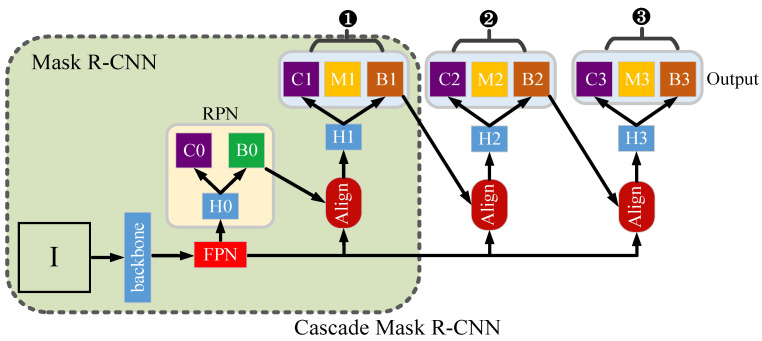
Structure diagram of Mask R-CNN and Cascade Mask R-CNN.

**Figure 3 animals-13-02181-f003:**
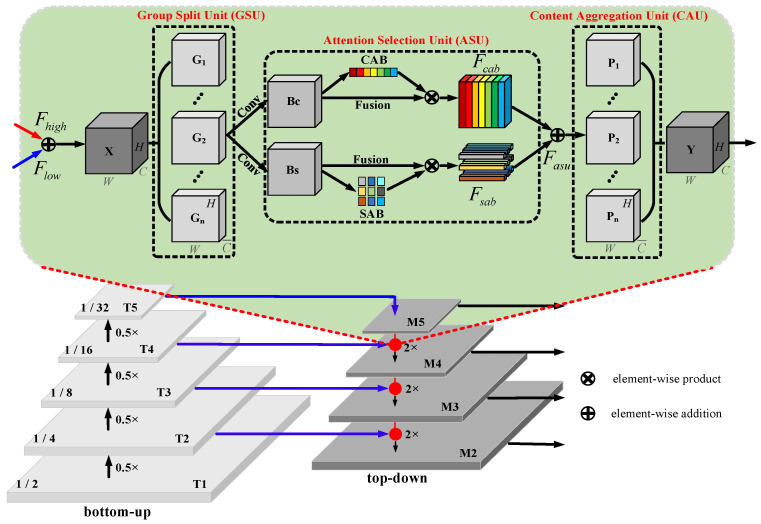
Feature pyramid network with group attention.

**Figure 4 animals-13-02181-f004:**
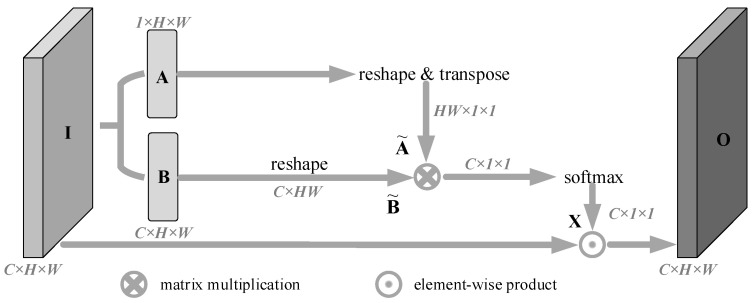
Channel attention branch.

**Figure 5 animals-13-02181-f005:**
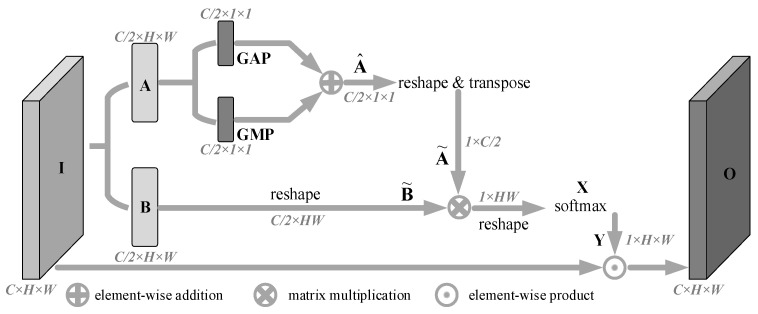
Spatial attention branch.

**Figure 6 animals-13-02181-f006:**
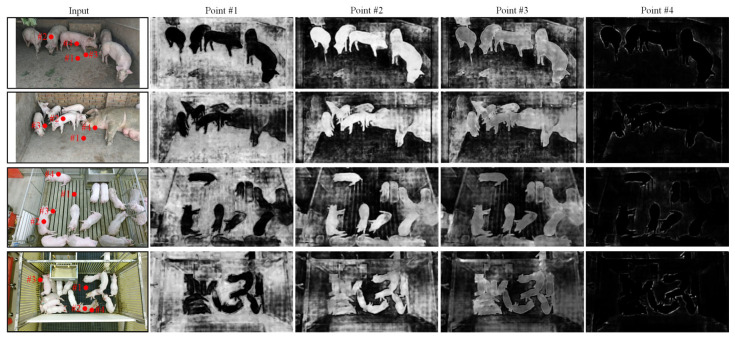
Partial feature maps filtered by spatial attention branch. Red dots indicate areas that need to be focused on.

**Figure 7 animals-13-02181-f007:**
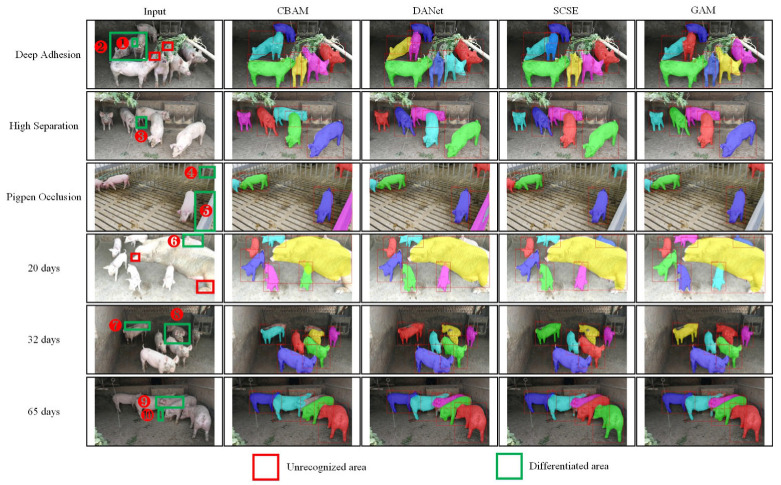
Visualization results of different scenarios and age stages. The circled numbers in the figure indicate the parts that need to be paid attention to.

**Figure 8 animals-13-02181-f008:**
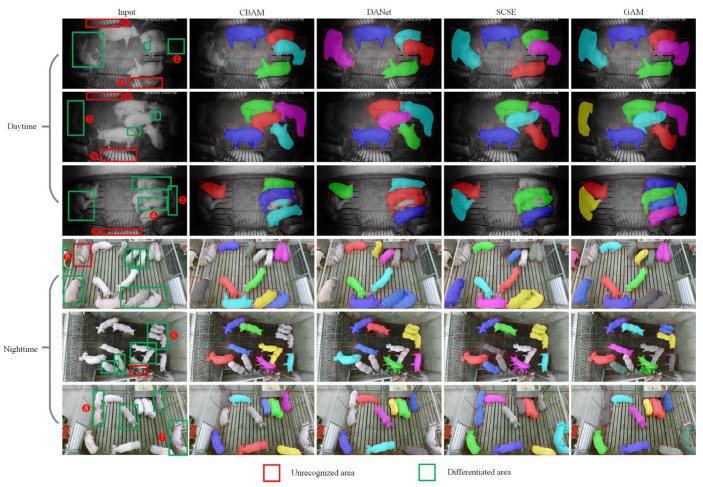
Day and night image prediction results of other datasets. The circled numbers in the figure indicate the parts that need to be paid attention to.

**Table 1 animals-13-02181-t001:** Glossary of important terms.

Abbreviation	Meaning
CNN	Convolutional Neural Network
FCN	Fully Convolutional Network
CBAM	Convolutional Block Attention Module
DANet	Dual Attention Network
SCSE	Concurrent Spatial and Channel Squeeze and Channel Excitation
Mask R-CNN	Mask Region-based Convolutional Neural Network
Cascade Mask R-CNN	Cascade Mask Region-based Convolutional Neural Network
ResNet	Residual Network
RPN	Region Proposal Network
FPN	Feature Pyramid Network
IOU	Intersection over Union
GAM	Group Attention Module
GSU	Group Split Unit
ASU	Attention Selection Unit
CAU	Content Aggregation Unit
CAB	Channel Attention Branch
SAB	Spatial Attention Branch
AP	Average Precision

**Table 2 animals-13-02181-t002:** Data collection environment information of different pig farms.

Farm Name	Collection Time	Collection Temperature	Collection Environment	Pigpen Size
Farm One	1 June 2019 9:00~14:00	Sunny, 23~29 °C	Outdoor, bright light	3.5 m × 2.5 m × 1 m
Farm Two	13 October 2019 10:30~12:00	Cloudy, 10~19 °C	Indoors, low light	4 m × 2.7 m × 1 m

**Table 3 animals-13-02181-t003:** Data augmentation operations.

Augmentation Method	Parameter Settings	Probability
Translation zoom and rotate	The translation factor is 0.0625, the image scaling and rotation factors are set to 0.1~0.3, and linear interpolation is used to fill the area where the translation occurs.	0.5
Randomly change brightness and contrast	The brightness and contrast variation range factors are both set to 0.1~0.3.	0.2
RGB value transformation	The R/G/B three-channel random transformation range is set to 0~10.	0.1
HSV value transformation	The range of H/S/V random transformation is set to 0~20, 0~30, and 0~20, respectively.	0.1
Image compression	The upper and lower limits of the compression percentage are set to 95 and 85, respectively.	0.2
Randomly rearrange channels	——	0.1
Median blur	The filter radius is set to 3.	0.1

**Table 4 animals-13-02181-t004:** Test results before and after the introduction of data enhancement and DCN during training.

Task Network	Backbone Attention Type	Extra	*AP* ^50^	*AP* ^75^	*AP* ^L^	*AP*
Mask R-CNN	0010	NONE	89.9	80.2	68.4	66.0
		Albu	91.4	80.9	68.8	66.5
		Albu-DCN	**91.8**	**82.3**	**69.7**	**67.3**
	1111	NONE	91.6	82.0	69.6	67.2
		Albu	91.9	83.1	70.1	67.7
		Albu-DCN	**92.2**	**83.6**	**70.6**	**68.1**
Cascade Mask R-CNN	0010	NONE	91.1	82.5	70.4	68.0
		Albu	91.7	82.9	70.6	68.3
		Albu-DCN	**91.9**	**83.0**	**70.7**	**68.5**
	1111	NONE	91.9	83.3	71.0	68.7
		Albu	**92.3**	83.1	71.2	68.8
		Albu-DCN	92.1	**83.6**	**71.4**	**69.0**

Note: 0010 and 1111, respectively, represent the two attention structures proposed by [35]; NONE represents not using training data enhancement strategy; Albu represents using training data enhancement strategy; DCN represents a deformable convolutional network. The bold part indicates the best precision value.

**Table 5 animals-13-02181-t005:** *AP* index values of different attention modules under the condition of adding different backbones.

Task Network	Extra	Attention Module	*AP* ^50^	*AP* ^75^	*AP* ^L^	*AP*
Mask R-CNN	0010-Albu-DCN	CBAM	92.1	82.5	69.9	67.5
		DANet	91.8	83.1	69.8	67.4
		SCSE	**92.3**	82.3	70.1	67.7
		GAM	**92.3**	**83.3**	**70.8**	**68.3**
	1111-Albu-DCN	CBAM	92.3	**84.4**	71.6	69.1
		DANet	92.7	84.0	71.7	69.2
		SCSE	92.6	84.0	72.1	69.6
		GAM	**93.3**	**84.7**	**72.7**	**70.3**
Cascade Mask R-CNN	0010-Albu-DCN	CBAM	93.0	83.9	72.1	69.6
		DANet	92.9	84.5	72.1	69.7
		SCSE	92.4	84.2	72.2	69.9
		GAM	**93.4**	**84.9**	**72.8**	**70.5**
	1111-Albu-DCN	CBAM	93.9	85.8	73.7	71.3
		DANet	94.2	86.2	73.9	71.5
		SCSE	**94.4**	86.4	74.2	71.9
		GAM	**94.4**	**86.5**	**74.6**	**72.2**

Note: The bold part indicates the best precision value.

**Table 6 animals-13-02181-t006:** The influence of the number of different groups on the prediction performance.

Task Network	Group Size	*AP* ^50^	*AP* ^75^	*AP* ^L^	*AP*
Mask R-CNN	8	92.1	83.6	71.4	68.9
	16	**93.3**	**84.7**	**72.7**	**70.3**
	32	92.5	84.1	72.0	69.5
	64	93.1	84.4	72.4	69.8
Cascade Mask R-CNN	8	93.9	86.3	74.1	71.7
	16	**94.4**	**86.5**	**74.6**	**72.2**
	32	93.7	85.9	73.9	71.6
	64	94.0	86.2	74.1	71.7

Note: The bold part indicates the best precision value.

**Table 7 animals-13-02181-t007:** The number of parameters and calculations after adding different attention modules, where the unit of calculation is GFLOPs, which means one billion floating-point operations.

Model	Parameters	GFLOPs
CBAM	+0.020 M	+0.030
DANet	+0.041 M	+0.034
SCSE	+0.033 M	+0.031
GAM	+0.038 M	+0.032

## Data Availability

Not applicable.

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
