# Peer review of "Attention-Guided Instance Segmentation for Group-Raised Pigs"

_animals, 2023, doi:10.3390/ani13132181_

Round 1
Reviewer 1 Report
In this study, it has been proposed a grouped attention module that combines channel attention and spatial attention simultaneously and apply it to a feature pyramid network for instance segmentation of group-raised pigs. Hereafter, my comments:
Section 3.2] It is not clear where GSU, ASU and CAU are placed in relation to Figure 2. May you clarify it and even provide an overall network picture?
Section 4.2] May you provide more details about "Average Precision (AP)"?
Section 4.3.1] May you clarify what "attention structures 0010 and 1111" are?
Author Response
Reply to Reviewer
Thanks for your time and efforts devoted to the review of our paper, which are much appreciated. All your comments have been addressed in this response.
General comments:
Comment: “In this study, it has been proposed a grouped attention module that combines channel attention and spatial attention simultaneously and apply it to a feature pyramid network for instance segmentation of group-raised pigs.”
Response: Thanks so much indeed for your positive comments.
Specific comments:
Comment: “Section 3.2] It is not clear where GSU, ASU and CAU are placed in relation to Figure 2. May you clarify it and even provide an overall network picture?”
Response: Thanks for your insightful comment. I have made modifications to Figure 2 by highlighting the content corresponding to GSU, ASU, and CAU in bold and changing the font color to red.
Comment: “Section 4.2] May you provide more details about "Average Precision (AP)"?”
Response: Thanks for your insightful comment. We have added the explanations for Equations (3) to (5) to clarify the AP metric. Additionally, we have included descriptions of the symbols used in Equations (3) to (5) in the paper. The corresponding added content is as follows:
“AP represents the area under the Precisioon-Recall curve, which can be shown in formulas (3)~(5), where TP (True Positive) represents the number of pixels correctly predicted as the pig category, FP (False Positive) denotes the number of pixels wrongly predicted as the pig category, and FN (False negative) represents the number of pixels predicted as the background instead of the pig category.”
Comment: “Section 4.3.1] May you clarify what "attention structures 0010 and 1111" are?”
Response: Thanks for your insightful comment. In Section 4.3.1, we introduced the following content to clarify what exactly is meant by 0010 and 1111.
In backbone networks, attention mechanisms can also be introduced, and the performance of the attention mechanism can be influenced by several factors, including the content of query and key, the content of query and relative position, only the content of key, and only the relative position. Among them, attention formed by simultaneously using all four types of content is referred to as "1111," while attention that only incorporates the content of key is referred to as "0010."
Reviewer 2 Report
This paper proposed new grouped attention module in instance segmentation of group raised pigs.
But the accuracy improvement is little in my view. The new structure design is so good,
but the complexity becomes too high. In this field, the dataset size is more
problematic for realistic improvement of deep learning technology, I think.
Specific comments
1. You made an additional deep learning architecture from ref 27. Dual attention was changed to grouped
attention. Augmentation methods newly were applied with ref 35. Cascaded Mask RCNN was also newly tested. But (AP0.5, AP0.75) improved from (Dual attention of ref27)(93.1%, 84.1%) to (Cascade Mask R-CNN, 1111-Albu-DCN)(94.4%, 86.5%). This can come from increase of system complexities.
2. In Table 4, there are so small differences between (CBAM, DANet and SCSE) and proposed method.
This experiments can decrease the interest of readers because of length of this paper.
3. Deformable convolution networks(DCN) appeared in section 4.3 without specific explanations. You must
provide the architecture figure of these DCN networks in my thoughts. It is difficult to understand your augmentation methods.
DCN comes from registration concepts, but your paper is on segmentation field. Too complex architecture was tried without a substantial accuracy improvement. These experiments can decrease the interest of readers because of length of this paper.
4. GPT4 and DALLE capture extreme attentions from world. You can add relationship discussion between your attention system and these systems in discussion section for interests of readers.
5. In line 272, the eigenvalue of the i-th row of B tilda was calculated by PCA(Principal Component Analysis)? You can add exact explanations.
6. You can add a table of (parameter numbers and FLOPs) under several tested systems for quality improvment.
7.Rectangles of "unrecognized area" and "differentiated area" in figure 7 and 8 are too vague.
Author Response
Reply to Reviewer
Thanks for your time and efforts devoted to the review of our paper, which are much appreciated. All your comments have been addressed in this response.
General comments:
Comment: “This paper proposed new grouped attention module in instance segmentation of group raised pigs. But the accuracy improvement is little in my view. The new structure design is so good, but the complexity becomes too high. In this field, the dataset size is more problematic for realistic improvement of deep learning technology.”
Response: Thanks so much indeed for your positive comments.
- Firstly, regarding the question about the modest improvement in accuracy, we would like to clarify that the attention-based module has a plug-and-play nature. Unlike introducing different depths of backbone networks, it requires fewer computational resources to achieve a certain degree of accuracy improvement. As mentioned by CBAM [1], DANet [2], and SCSE [3], the magnitude of this improvement is not expected to be significant. Therefore, the accuracy improvement presented in the paper is within a reasonable range. We would like to express our gratitude to the reviewing professor for their valuable suggestions.
- Then, regarding the computational cost issue introduced by the attention mechanism, it is undeniable that incorporating attention does come with a certain computational overhead compared to not having attention. However, this overhead is tolerable and not as significant as modifying the backbone network, which would introduce a larger number of parameters. Additionally, it is important to note that we are further improving the model's performance on an already high-accuracy backbone network. Therefore, the core reason for introducing the attention mechanism is to continue enhancing the accuracy under the condition of an already high level of performance, which inherently limits the extent of improvement that can be achieved.
- Regarding the scale of the dataset, it is currently limited due to the need for data collection, manual annotation, and verification processes. However, our dataset is continuously expanding. It includes various aspects such as different pig ages, diverse capture angles, different capture times, and varying lighting conditions. In the future, we plan to open-source our dataset and organize large-scale data science competitions to promote the development of this field.
[1] Woo, S.Y.; Park, J.C.; Lee, J.Y.; Kweon, I.S. Cbam: Convolutional block attention module. ECCV. 2018, 3-19.
[2] Fu, J.; Liu, J.; Tian, H.J.; Li, Y.; Bao, Y.J.; Fang, Z.W.; Lu, H.Q. Dual attention network for scene segmentation. CVPR. 2019, 3146-3154.
[3] Roy, A.G.; Navab, N.; Wachinger, C. Concurrent spatial and channel squeeze & excitation in fully convolutional networks. MICCAI. 2018, 421-429.
Specific comments:
Comment: “You made an additional deep learning architecture from ref 27. Dual attention was changed to grouped attention. Augmentation methods newly were applied with ref 35. Cascaded Mask RCNN was also newly tested. But (AP0.5, AP0.75) improved from (Dual attention of ref27)(93.1%, 84.1%) to (Cascade Mask R-CNN, 1111-Albu-DCN)(94.4%, 86.5%). This can come from increase of system complexities.”
Response: Thanks for your insightful comment. We have applied group attention in the feature pyramid network, but it is important to note that the model's performance is not solely determined by the number of parameters or the complexity of the model. Introducing too many modules can sometimes lead to overfitting issues. Therefore, the increase in performance should not be directly associated with an increase in complexity. If we were to make a comparison, it is true that incorporating attention introduces additional complexity compared to not using attention. However, this introduction is worthwhile and unavoidable because achieving higher model accuracy often requires sacrificing some speed. This is my personal viewpoint, and I look forward to further discussing it with the reviewing professor.
Comment: “In Table 4, there are so small differences between (CBAM, DANet and SCSE) and proposed method. This experiments can decrease the interest of readers because of length of this paper.”
Response: Thanks for your insightful comment. The reviewing professor is correct in stating that, from a numerical perspective, the improvement of our model compared to CBAM, DANet, and SCSE is not significant. However, it should be noted that our main contribution lies in introducing the group attention mechanism and applying it to the field of pig instance segmentation. Personally, I believe that proposing a model that achieves comparable or even superior performance to existing models by adopting different modules is also a form of innovation. In our future research, we will continue refining our model based on the professor's suggestions. This includes exploring ways to enhance the rationality of attention settings and improve the effectiveness of the attention mechanism.
Comment: “Deformable convolution networks(DCN) appeared in section 4.3 without specific explanations. You must provide the architecture figure of these DCN networks in my thoughts. It is difficult to understand your augmentation methods. DCN comes from registration concepts, but your paper is on segmentation field. Too complex architecture was tried without a substantial accuracy improvement. These experiments can decrease the interest of readers because of length of this paper.”
Response: Thanks for your insightful comment. DCN is a commonly used technique in deep learning. It is employed to replace conventional convolution operations. However, since the focus of this article is not to introduce DCN, we have only utilized it in the ResNet backbone network without providing an extensive explanation of DCN. Nevertheless, in accordance with the suggestions from the reviewer, we have added the following content to briefly describe DCN in the original text.
DCN is an improved operations on the basis of traditional CNNs, by making small offsets to the sampling positions, it enables non-linear deformation modeling of input features. In image segmentation, it can provide more accurate position and shape information. We replaced the regular convolutions in ResNet50 with DCN to improve the accuracy of the segmentation results.
Regarding the data augmentation approach, we did not employ the method of augmenting the entire dataset first and then conducting experiments on the augmented dataset. Instead, we adopted an on-the-fly data augmentation strategy during training. For each batch of data, we randomly select augmentation operations to expand the dataset. This approach enhances the diversity of data variations. It has been proven effective in several data science competitions.
Comment: “GPT4 and DALLE capture extreme attentions from world. You can add relationship discussion between your attention system and these systems in discussion section for interests of readers.”
Response: Thanks for your insightful comment. I have added the following paragraph in the discussion section of the document. Thank you very much, this discussion section feels necessary and aligned with the forefront of research.
In addition, GPT-4 and DALL·E models, as two generative models based on the Transformer architecture, can have two potential applications in the field of pig instance segmentation for dataset augmentation. On the one hand, GPT-4 can be utilized for instance-level annotation of pig image data. Since GPT-4 supports multimodal operations, it can provide coarse-grained annotations, which can then be further refined through manual verification. On the other hand, with the help of the DALL·E model, more complex pig datasets can be generated based on natural language descriptions. The utilization of these two technologies is expected to bring innovation to the field of pig instance segmentation.
Comment: “In line 272, the eigenvalue of the i-th row of B tilda was calculated by PCA(Principal Component Analysis)? You can add exact explanations.”
Response: Thanks for your insightful comment. No, the term eigenvalue here refers to the value in the feature maps and is different from the eigenvalues used in PCA dimensionality reduction algorithms. In order to avoid confusion, I have modified the description of "eigenvalue" to "value" . Thank you for pointing that out.
Comment: “You can add a table of (parameter numbers and FLOPs) under several tested systems for quality improvement.”
Response: Thanks for your insightful comment. Indeed, we lack a comparison table between model parameter count and computational complexity. Therefore, we have added the following content to the revised draft:
In deep learning, model parameter count and computational complexity are two important indicators used to measure the complexity of a model and its computational resource requirements. Model parameter count refers to the number of adjustable parameters that need to be learned in the model. These parameters are used to represent the weights and biases of the model, and by adjusting them, the model can adapt to the given training data. A larger parameter count generally indicates a stronger representation capacity of the model, but it also increases the model's complexity and memory consumption. Computational complexity refers to the total number of computational operations required during model inference or training. Computational complexity is typically related to the model's structure and the size of the input data. In deep learning, computational complexity is often measured in terms of the number of floating-point operations (FLOPs) or the count of multiplication-addition operations. A higher computational complexity implies the need for more computational resources (such as CPUs or GPUs) to perform the model's forward and backward propagation, thereby increasing the computation time and cost. The parameters and calculation results before and after adding different attention are shown in Table 7. It can be found that the introduction of attention does bring a certain amount of time and space overhead, but this overhead is relatively reasonable and tolerable.
Model |
Parameters |
GFLOPs |
CBAM |
+0.020M |
+0.030 |
DANet |
+0.041M |
+0.034 |
SCSE |
+0.033M |
+0.031 |
GAM |
+0.038M |
+0.032 |
Comment: “Rectangles of "unrecognized area" and "differentiated area" in figure 7 and 8 are too vague.”
Response: Thanks for your insightful comment. Regarding Figures 7 and 8, the main reason for the blurry rectangular boxes in the manuscript is due to the file size limitations when submitting the manuscript. As a result, Figures 7 and 8 underwent compression during the submission process. However, once the manuscript is accepted, we will provide the original high-resolution images to the editorial office. Thank you for bringing this to our attention.
Reviewer 3 Report
This manuscript aimed at validating the effectiveness of attention mechanisms in instance segmentation tasks for group-raised pigs. The deep learning methodology was clearly presented and demonstrated. However, the novelty of the study was limited to the domain of livestock farming, as there already exist segmentation techniques for pigs, and similar methods have been applied to other livestock species. The authors may consider improving the manuscript by emphasizing its impacts on practical livestock farming and/or coupling the proposed method with a downstream computer vision task e.g., identification to show the value of the work that fills gaps in the livestock sector. Otherwise, the current manuscript seems to be another deep learning paper in this field. Below are my comments.
Simple summary
The authors may improve this section by avoiding technical terms e.g., channel attention, spatial attention, feature pyramid network, etc. Considering the readership of the journal, the current form may be abrupt.
Abstract
Lines 17-19: did the authors mean segmentation in computer vision?
This sentence seemed to be disconnected from the previous and next sentences.
Line 29: the 0010 and 1111 attention information was not explained previously, and the sentence was hard to follow.
The abstract section needs to be rewritten. There is a lack of logics, and the sentences do not flow well. More descriptions of experimental design are encouraged. The authors presented a set of technical terms without definition/explanation which negatively affects reading. No core results were shown and thus conclusions were not backed up.
Introduction
Lines 47-55: I strongly recommend rewriting this paragraph. There are logical problems. For instance, I have not heard about the term “free-range group breeding” and furthermore, breeding cannot be the main feeding method. Moreover, which nation did the authors refer to when talking about food security? If a specific nation was referred to, why there was no background introduction about food security in the referred nation? There was no citation throughout the paragraph, and I was not sure whether the authors were speculating by making arguments or using known facts.
The introduction section was heavy on deep learning technical details while less attention was paid to its contributions to swine farming. What could this work bring to the livestock sector besides experimenting on a new deep learning structure? There already exist many deep-learning segmenting methods for pigs. How does the proposed work outstand in terms of generalizability and robustness?
Materials and Methods
Data collection: the justification to use the horizontal view of the camera was not convincing. First, for applied values, the majority of pig and pig behavior recognition systems utilize commercial cameras with fixed lens in top-down views. How the high-end camera used in the study can be applied in scale and in production? Second, how face and hoof information is considered more valuable for identification if the pigs have back marks or ear tags? Further, top-down views are usually considered occlusion-free as in fewer cases there are objects between the camera and the pigs. Lastly, there are evidence and papers showing that top-down view computer vision systems help recognize aggressive behaviors of pigs with high accuracy. How do side-view images/videos bring additional value to aggression detection?
Data processing: how was the test set sampled? In machine learning test set is typically considered independent of the training set. Did the authors reflect on the point?
The main concern with the current form is the flow of reading. Sentences are disconnected and could be abrupt. In addition, there are several technical terms introduced without being defined previously.
Author Response
Reply to Reviewer
Thanks for your time and efforts devoted to the review of our paper, which are much appreciated. All your comments have been addressed in this response.
General comments:
Comment: “The authors may improve this section by avoiding technical terms e.g., channel attention, spatial attention, feature pyramid network, etc. Considering the readership of the journal, the current form may be abrupt.”
Abbreviation |
Meaning |
CNN |
Convolutional Neural Network |
FCN |
Fully Convolutional Network |
CBAM |
Convolutional Block Attention Module |
DANet |
Dual Attention Network |
SCSE |
Concurrent Spatial and Channel Squeeze and Channel Excitation |
Mask R-CNN |
Mask Region-based Convolutional Neural Network |
Cascade Mask R-CNN |
Cascade Mask Region-based Convolutional Neural Network |
ResNet |
Residual Network |
RPN |
Region Proposal Network |
FPN |
Feature Pyramid Network |
IOU |
Intersection over Union |
GAM |
Group Attention Module |
GSU |
Group Split Unit |
ASU |
Attention Selection Unit |
CAU |
Content Aggregation Unit |
CAB |
Channel Attention Branch |
SAB |
Spatial Attention Branch |
AP |
Average Precision |
Response: Thanks so much indeed for your positive comments. In order to make it easier for readers to refer to the terms that appear in the article, we extracted some terms that appear frequently in the text to form a glossary, and placed it at the very beginning of the manuscript.
Specific comments:
Abstract:
Comment: “Lines 17-19: did the authors mean segmentation in computer vision? This sentence seemed to be disconnected from the previous and next sentences.”
Response: Thanks for your insightful comment. Yes, referring to segmentation here, we have rewritten the summary section.
Comment: “Line 29: the 0010 and 1111 attention information was not explained previously, and the sentence was hard to follow. The abstract section needs to be rewritten. There is a lack of logics, and the sentences do not flow well. More descriptions of experimental design are encouraged. The authors presented a set of technical terms without definition/explanation which negatively affects reading. No core results were shown and thus conclusions were not backed up.”
Response: Thanks for your insightful comment. We have modified the abstract part of the original text, the corresponding content is as follows, the modified part of the original text can be seen in the red font part:
“In the pig farming environment, complex factors such as pig adhesion, occlusion, and changes in body posture pose significant challenges for segmenting multiple target pigs. To address these challenges, this study collected video data using a horizontal angle of view and a non-fixed lens. Specifically, a total of 45 pigs aged 20 to 105 days in 8 pens were selected as research subjects, resulting in 1917 labeled images. These images were divided into 959 for training, 192 for validation, and 766 for testing. The grouped attention module was employed in the feature pyramid network to fuse the feature maps from deep and shallow layers. The grouped attention module consists of a channel attention branch and a spatial attention branch. The channel attention branch effectively models dependencies between channels to enhance feature mapping between related channels and improve semantic feature representation. The spatial attention branch establishes pixel-level dependencies by using the response values of all pixels in a single-channel feature map to the target pixel. It further guides the original feature map to filter spatial location information and generate context-related outputs. The grouped attention, along with data augmentation strategies, was incorporated into the Mask R-CNN and Cascade Mask R-CNN task networks to explore their impact on pig segmentation. The experiments showed that introducing data augmentation strategies improved the segmentation performance of the model to a certain extent. Taking Mask-RCNN as an example, under the same experimental conditions, the introduction of data augmentation strategies resulted in improvements of 1.5%, 0.7%, 0.4% and 0.5% in metrics AP50, AP75, APL, and AP, respectively. Furthermore, our grouped attention module achieved the best performance. For example, compared to existing attention module CBAM, taking Mask R-CNN as an example, in terms of metric AP50, AP75, APL and AP, the grouped attention outperformed 1.0%, 0.3%, 1.1% and 1.2%, respectively. We further studied the impact of the number of groups in the grouped attention on the final segmentation results. Additionally, visualizations of predictions on third-party data collected using a top-down data acquisition method, which were not involved in the model training, demonstrated that the proposed model in this paper still achieved good segmentation results, proving the transferability and robustness of the grouped attention. Through comprehensive analysis, we found that grouped attention is beneficial for achieving high-precision segmentation of individual pigs in different scenes, ages, and time periods. The research results can provide references for subsequent applications such as pig identification and behavior analysis in mobile settings.”
Introduction:
Comment: “Lines 47-55: I strongly recommend rewriting this paragraph. There are logical problems. For instance, I have not heard about the term “free-range group breeding” and furthermore, breeding cannot be the main feeding method. Moreover, which nation did the authors refer to when talking about food security? If a specific nation was referred to, why there was no background introduction about food security in the referred nation? There was no citation throughout the paragraph, and I was not sure whether the authors were speculating by making arguments or using known facts.”
Response: Thanks for your insightful comment. I have rewritten Lines 47-55 as the following passage:
“With increasing concerns about food safety and health, the pork industry is placing greater emphasis on pig breeding and management. However, outbreaks of contagious diseases such as African swine fever and increasing environmental pressures have posed many challenges to the pig farming industry. To improve pig farming efficiency and health, monitoring and managing the health status of pigs has become crucial. Traditional methods of monitoring pig health mainly rely on manual observation, but this approach is not only time-consuming and labor-intensive, but also susceptible to subjective factors and has a certain misjudgment rate. As pig farming is moving towards intensification, scale, and facility-based management, fine pig farming based on individual management and quality assurance that meets animal welfare requirements has become the trend in pig farming industry. Instance segmentation technology can automatically analyze and recognize various features of pigs by capturing video or image data of pigs in their natural state, and provide diagnosis results. This method not only reduces labor costs but also can more accurately identify and diagnose the health status of pigs, thereby improving pig farming efficiency and health.”
Comment: “The introduction section was heavy on deep learning technical details while less attention was paid to its contributions to swine farming. What could this work bring to the livestock sector besides experimenting on a new deep learning structure? There already exist many deep-learning segmenting methods for pigs. How does the proposed work outstand in terms of generalizability and robustness?”
Response: Thanks for your insightful comment.
- Regard “What could this work bring to the livestock sector besides experimenting on a new deep learning structure?”
This work advances the application of attention mechanisms in the field of pig instance segmentation. It further demonstrates the effectiveness of attention mechanisms for pig segmentation through multiple experimental validations. Additionally, this work is conducted using a dataset from a horizontal perspective, deviating from the majority of works that acquire data from a top-down approach. This is beneficial for future research to be conducted on multi-view angle datasets. Lastly, we visualize the attention maps and validate the transferability and robustness of the model on top-down views. These are some of the contributions achieved in this research work.
- Regard “How does the proposed work outstand in terms of generalizability and robustness?”
Although the model training in this work was conducted on a dataset from a horizontal perspective, Figure 8 of the article shows that we also tested the model on top-down view data, which was not involved in the training process. This even included data from nighttime scenes. From the prediction results, it is evident that our model performs well in accurately segmenting individual pigs. These findings provide strong evidence for the robustness and transferability of our model.
Materials and Methods:
Comment: “Data collection: the justification to use the horizontal view of the camera was not convincing. First, for applied values, the majority of pig and pig behavior recognition systems utilize commercial cameras with fixed lens in top-down views. How the high-end camera used in the study can be applied in scale and in production? Second, how face and hoof information is considered more valuable for identification if the pigs have back marks or ear tags? Further, top-down views are usually considered occlusion-free as in fewer cases there are objects between the camera and the pigs. Lastly, there are evidence and papers showing that top-down view computer vision systems help recognize aggressive behaviors of pigs with high accuracy. How do side-view images/videos bring additional value to aggression detection?”
Response: Thanks for your insightful comment.
- Regard “First, for applied values, the majority of pig and pig behavior recognition systems utilize commercial cameras with fixed lens in top-down views. How the high-end camera used in the study can be applied in scale and in production?”
We acknowledge the use of a top-down approach for studying individual pigs. However, in this article, we explore a new perspective on data collection that we believe is valuable. There are several factors to consider: Firstly, the horizontal perspective has not been extensively studied by researchers in this field, making it an exploratory direction worth pursuing. Secondly, using a horizontal perspective allows us to gather data that captures certain characteristics not obtainable through the top-down approach. For instance, the horizontal perspective can capture details such as the pig's face, hooves, and other finer aspects, whereas the top-down approach primarily focuses on the back information. Lastly, models trained on images obtained from the horizontal perspective may be better suited for deployment on mobile devices, considering the practical applications and limitations of such devices.
- Regard “Second, how face and hoof information is considered more valuable for identification if the pigs have back marks or ear tags? Further, top-down views are usually considered occlusion-free as in fewer cases there are objects between the camera and the pigs”
Yes, you are correct. In the top-down data collection approach, there is no obstruction between the camera and the pigs. In my opinion, both the top-down and horizontal perspective data collection methods have their own advantages and disadvantages. Regardless of the scenario, both types of data are worth researching. You have already collected some top-down experimental data, and your ultimate goal is to train a multi-view, all-weather pig instance segmentation model that can adapt to data from multiple sources. Because the horizontal perspective does have practical significance in actual production, you chose horizontal data for this article. However, in our next research work, we will incorporate top-down views as well. Your suggestions are highly valuable.
- Regard “Lastly, there are evidence and papers showing that top-down view computer vision systems help recognize aggressive behaviors of pigs with high accuracy. How do side-view images/videos bring additional value to aggression detection?”
Regarding the attacking behavior of pigs, such as climbing or crossing, personally, I believe that the horizontal perspective may be better at capturing this type of information. However, this is just my personal viewpoint, and it may conflict with yours. As I mentioned earlier, there is no absolute superiority or inferiority between the horizontal and top-down perspectives. They are simply different viewpoints for observing pigs. If the data collected from both perspectives can be used to train a unified multi-view model in subsequent stages, then this issue will no longer exist. This is indeed our next task, and the reviewer has provided us with a great starting point for future research. Thank you for your valuable input.
Comment: “Data processing: how was the test set sampled? In machine learning test set is typically considered independent of the training set. Did the authors reflect on the point?”
Response: Thanks for your insightful comment. Our training set, test set, and validation set are all randomly extracted from the entire dataset. This approach has two advantages: on one hand, it ensures the consistency of the dataset distribution because each image is sampled with the same probability. On the other hand, it guarantees that there is no data overlap between the training set, validation set, and test set. Yes, as the reviewing professor mentioned, it is absolutely not allowed for the test set to have any overlap with the training set.

Round 2
Reviewer 2 Report
According to my comments, authors added a complexity comparison table and additional explanations of DCN. Contents of Abstract section are corrected by clearing the accuracy improvment for easy reading. Furthermore, authors insisted the needs of research in deep learning architecture, although accuracy improvment is not substantial. Deep learning architecture is not mature in instance-segmentation field. This paper is valuable in this respect in my thoughts.
Author Response
Reply to Reviewer
Thanks for your time and efforts devoted to the review of our paper, which are much appreciated. All your comments have been addressed in this response.
General comments:
Comment: “According to my comments, authors added a complexity comparison table and additional explanations of DCN. Contents of Abstract section are corrected by clearing the accuracy improvement for easy reading. Furthermore, authors insisted the needs of research in deep learning architecture, although accuracy improvement is not substantial. Deep learning architecture is not mature in instance-segmentation field. This paper is valuable in this respect in my thoughts..”
Response: Thanks so much indeed for your positive comments.
Reviewer 3 Report
It is notable that the authors reflected on my comments made for the previous version. However, the revised manuscript did not completely address my concerns, and I would suggest another round of revision. Note that the authors presented the deep learning methodology in an excellent way, and I have no problem with the technical details related to the attention-based segmentation models. My comments are centered around the impact of the work on practical swine farming. I need to point out that this manuscript seemed to be written for the readership of the computer science & engineering and/or data science community, but it would be quite challenging for animal scientists to get fully engaged. I’ve reviewed the authors' reply to my previous comments, and I encourage the authors to revisit the following points:
Introduction: my intent was to ask the authors to reflect on the mentioned impacts in writing. After reading the introduction my understanding wouldn’t change much unless I check the authors’ reply. What would be highlights that will draw attention from the readers with animal science background?
Material and methods: the horizontal camera view absolutely brings value for research purposes. However, if the authors are aware of pig pen designs in large-scale pig farming, it’s challenging to install and maintain the camera. Typically there is little space between pens, and fences and gates usually occlude horizontal camera views. Even if the camera is installed within the pen, cameras/lenses would be soon contaminated due to the harsh environment, while top-down cameras would be easier to use in terms of maintenance. Not sure what the authors mean by using mobile devices. Would that be suitable for continuous monitoring anymore?
Lastly, this work was built on a small dataset and unlike the benchmark studies with reference datasets e.g., ImageNet and COCO, the accuracy improvement may be limited or not fully representative. But the authors showed a good example of the model applied to a third-party dataset, which can be considered a test set that is independent of the training set and implies model generalizability. But I was not sure of the model performance/metrics on the independent data. To me, this work is more about showcasing modeling and hyperparameter tuning techniques on a small dataset that are not necessarily comparable to peers’ work, although I understand the pig farming sector has its own limitations on the reference image data. Again, the authors did a good job on the modeling, but need to provide more evidence to animal scientists to show its potential.
Minor comments:
Lines 59-61: how can instance segmentation provide diagnosis results? Wouldn’t that be coupled with other techniques e.g., classification or behavior detection models for diagnostic purposes? In addition, the citation indexing here was odd (the previous one was [1] and this one was [27])
Good luck!
Author Response
Reply to Reviewer
Thanks for your time and efforts devoted to the review of our paper, which are much appreciated. All your comments have been addressed in this response.
General comments:
Comment: “It is notable that the authors reflected on my comments made for the previous version. However, the revised manuscript did not completely address my concerns, and I would suggest another round of revision. Note that the authors presented the deep learning methodology in an excellent way, and I have no problem with the technical details related to the attention-based segmentation models. My comments are centered around the impact of the work on practical swine farming. I need to point out that this manuscript seemed to be written for the readership of the computer science & engineering and/or data science community, but it would be quite challenging for animal scientists to get fully engaged. I’ve reviewed the authors' reply to my previous comments, and I encourage the authors to revisit the following points:”
Response: Thanks so much indeed for your positive comments.
Specific comments:
Abstract:
Comment: “Introduction: my intent was to ask the authors to reflect on the mentioned impacts in writing. After reading the introduction my understanding wouldn’t change much unless I check the authors’ reply. What would be highlights that will draw attention from the readers with animal science background?”
Response: Thanks for your insightful comment. For individuals with a background in animal science, pig instance segmentation can bring the following benefits:
- Study of behavior and activity patterns: Pig instance segmentation provides precise contour information for each pig, allowing researchers to better observe and analyze pig behavior and activity patterns. Through quantitative analysis of pig movement trajectories and behaviors in specific environments, it is possible to gain deeper insights into pig habits and behavioral characteristics, providing scientific evidence for pig rearing management and welfare improvement.
- Health monitoring and disease diagnosis: Pig instance segmentation can assist researchers in monitoring the health and diagnosing diseases of each pig. By analyzing characteristics such as pig body posture, skin color, eyes, and nostrils, health issues in pigs can be promptly detected, enabling targeted treatment and management. Furthermore, by performing instance segmentation of pigs in different disease states, the impact of diseases on pig body shape and behavior can be studied, providing references for disease prevention and control.
- Rearing management and productivity improvement: Pig instance segmentation can help farmers to manage individual pigs more effectively. By performing instance segmentation and tracking of pigs, it is possible to monitor the growth and development, weight changes, and adaptability to rearing environments of pigs in real-time, allowing for timely adjustments to rearing strategies and improving rearing efficiency and productivity. Additionally, instance segmentation of pigs can be used for measuring and evaluating food intake, aiding in the proper control of feed distribution and pig dietary management.
In summary, pig instance segmentation offers animal science professionals the opportunity to delve into the study of pig behavior, health monitoring, and rearing management. It provides a scientific basis and technical support for pig welfare improvement, disease prevention, and productivity enhancement.
Comment: “Material and methods: the horizontal camera view absolutely brings value for research purposes. However, if the authors are aware of pig pen designs in large-scale pig farming, it’s challenging to install and maintain the camera. Typically there is little space between pens, and fences and gates usually occlude horizontal camera views. Even if the camera is installed within the pen, cameras/lenses would be soon contaminated due to the harsh environment, while top-down cameras would be easier to use in terms of maintenance. Not sure what the authors mean by using mobile devices. Would that be suitable for continuous monitoring anymore?”
Response: Thanks for your insightful comment. Yes, I agree with your point that installing a horizontal camera presents more challenges compared to a top-down installation method. However, as described in our original text, images obtained using a horizontal approach are better suited for mobile applications. The data in this study was collected using a camera in a horizontal position. Our ultimate goal is to embed the model from this study into a mobile application for smartphones. Since humans observe pigs from a horizontal perspective, a model trained on data obtained in a horizontal manner will better align with the angle at which humans observe things. This is why we chose the horizontal approach.
Comment: “Lastly, this work was built on a small dataset and unlike the benchmark studies with reference datasets e.g., ImageNet and COCO, the accuracy improvement may be limited or not fully representative. But the authors showed a good example of the model applied to a third-party dataset, which can be considered a test set that is independent of the training set and implies model generalizability. But I was not sure of the model performance/metrics on the independent data. To me, this work is more about showcasing modeling and hyperparameter tuning techniques on a small dataset that are not necessarily comparable to peers’ work, although I understand the pig farming sector has its own limitations on the reference image data. Again, the authors did a good job on the modeling, but need to provide more evidence to animal scientists to show its potential.”
Response: Thanks for your insightful comment. Yes, the dataset used in this work is indeed smaller compared to COCO. However, it is sufficient for training a deep learning model because we do not need to train the model from scratch. Instead, we only need to fine-tune the weights of a pre-trained model, which is already performing well. Therefore, in terms of data size, our dataset is currently sufficient. As you mentioned, we conducted prediction and visualization analysis of our model on a third-party dataset to validate its transferability and robustness. In practice, it is not possible to cover all data and scenarios. If a model exhibits good transferability, a model trained on a smaller dataset can easily adapt to a larger dataset. Furthermore, our research on pig instance segmentation is ongoing, and we are continuously expanding the dataset's size. This includes incorporating data from multiple sources, such as top-down data, different lighting conditions, different temperatures, different age groups of pigs, and more. We also plan to release the final version of the dataset for evaluation as an open-source resource. Therefore, the work presented in this article is part of our ongoing efforts in this field, and we will continue to update the dataset. Thank you for the reviewer's suggestions.
Comment: “Lines 59-61: how can instance segmentation provide diagnosis results? Wouldn’t that be coupled with other techniques e.g., classification or behavior detection models for diagnostic purposes? In addition, the citation indexing here was odd (the previous one was [1] and this one was [27])”
Response: Thanks for your insightful comment. In my opinion, pig instance segmentation is mainly applied in scenarios that require precise segmentation of pigs, such as pig disease detection and determination of the timing of pig delivery. These diagnoses require attention to the relatively important regions of the pig's body. Pig classification, on the other hand, is primarily used to analyze the breed and gender of pigs. Pig detection can be used for pig localization and tracking, as well as applications that require counting the number of pigs. Therefore, pig instance segmentation has its unique application scenarios. In addition, I have revised the numbering of the article references